# The Role of the TLR4-MyD88 Signaling Pathway in the Immune Response of the Selected Scallop Strain “Hongmo No. 1” to Heat Stress

**DOI:** 10.3390/ani14030497

**Published:** 2024-02-02

**Authors:** Chenyang Yue, Kexin Zhang, Zhigang Liu, Wengang Lü, Hui Guo, Liqiang Zhao, Xinyu Song, James Kar-Hei Fang

**Affiliations:** 1Guangdong Marine Invertebrates Science and Technology Innovation Center, Fisheries College, Guangdong Ocean University, Zhanjiang 524088, China; 2112001058@stu.gdou.edu.cn (K.Z.); lwg1125@gdou.edu.cn (W.L.); guoh@gdou.edu.cn (H.G.); lzhao@gdou.edu.cn (L.Z.); 2112101081@stu.gdou.edu.cn (X.S.); 2Guangdong Provincial Engineering Laboratory for Mariculture Organism Breeding, Fisheries College, Guangdong Ocean University, Zhanjiang 524088, China; 3Key Laboratory of Marine Ecology and Aquaculture Environment of Zhanjiang, Fisheries College, Guangdong Ocean University, Zhanjiang 524088, China; 4Department of Food Science and Nutrition, The Hong Kong Polytechnic University, Hong Kong 999077, China; james.fang@polyu.edu.hk

**Keywords:** TLR4-MyD88 signaling pathway, immune response, heat stress, scallops

## Abstract

**Simple Summary:**

The innate immunity of marine bivalves is challenged upon exposure to heat stress, but there is limited knowledge of its molecular mechanism. In this study, genes in the TLR4-MyD88 signaling pathway were investigated in a new scallop strain: “Hongmo No. 1” (Bohai Red ♂ × *A. irradians concentricus* ♀). This is the first study reporting that acute heat stress obviously inhibited the TLR4-MyD88 signaling pathway of bivalves, even with LPS stimulation. The subsequent RNAi targeting TLR4 gene in the new scallop strain “Hongmo No. 1” indicated the key role of TLR4-MyD88 signal transduction in the immune response. Acute heat stress seemed to affect the response of downstream mediator molecules to the inhibition of *HMTLR4*. These results provide more information about bivalve immunity under heat stress, which is expected to improve heat tolerance to cope with more frequent and extreme heat wave events.

**Abstract:**

The innate immunity of marine bivalves is challenged upon exposure to heat stress, especially with increases in the frequency and intensity of heat waves. TLR4 serves a classical pattern recognition receptor in recognizing pathogenic microorganisms and activating immune responses. In this study, three genes, *HMTLR4*, *HMMyD88* and *HMTRAF6*, were characterized as homologs of genes in the TLR4-MyD88 signaling pathway in the selected scallop strain “Hongmo No. 1”. According to RT-PCR, acute heat stress (32 °C) inhibited genes in the TLR4-MyD88 signaling pathway, and LPS stimulation-induced activation of TLR4-MyD88 signal transduction was also negatively affected at 32 °C. ELISA showed LPS-induced tumor necrosis factor alpha (TNF-α) or lysozyme (LZM) activity, but this was independent of temperature. RNA interference (RNAi) confirmed that *HMTLR4* silencing suppressed the expression of its downstream gene, whether at 24 °C or at 32 °C. The level of TNF-α and the activity of LZM also decreased after injection with dsRNA, indicating a negative effect on the innate immunity of scallops. Additionally, acute heat stress affected the suppression of downstream gene expression when compared with that at 24 °C, which led us to the hypothesis that heat stress directly influences the downstream targets of *HMTLR4*. These results enrich the knowledge of scallop immunity under heat stress and can be beneficial for the genetic improvement of new scallop strains with higher thermotolerance.

## 1. Introduction

The global ocean continues to experience the consequences of increases in the frequency and intensity of heat waves [1], and acute heat stress poses a significant threat to biodiversity, including economically important marine organisms, especially species with a limited ability to move [2]. Marine bivalves are poikilotherms whose body temperature changes as the ambient water temperature changes, which means they are vulnerable to harm from acute heat stress. Various physiological and biochemical changes have been observed in bivalves under heat stress, such as negatively correlated growth, increased energy metabolism and weakened or strengthened immunity [3,4,5].

The bay scallop *Argopecten irradians concentricus* was originally introduced from the Atlantic coast of America to China and cultured in Guangdong Province in the 1990s. Wang et al. [6] crossed *A. irradians concentricus* and *Argopecten purpuratus*, continued the selection of offspring by population breeding, and produced a new strain called Bohai Red (a scallop variety in China, variety registration no. GS-01-003-2015) with faster growth but relatively low heat stress tolerance (semi-lethal temperature 29 °C for 96 h) [7]. For further genetic improvement with environmental adaption to high temperatures in subtropical areas, a newly selected strain of scallop, “Hongmo No. 1” (Bohai Red ♂ × *A. irradians concentricus* ♀), was produced, and this strain exhibited higher heat stress tolerance (semi-lethal temperature 32.4 °C for 96 h) [8]. Despite the genetic improvement, more frequent and extreme heat wave events still cause scallop cultures in South China acute stress. Further understanding of the response of scallops to heat stress is required for the future genetic improvement of the “Hongmo No. 1” strain. The immune response has received a great deal of research attention for its close relationship with heat stress-induced mortality.

The innate immunity of bivalves is their primary line of defense, including at humoral and cellular levels. Immunological recognition is regarded as the initial step of effective defense, during which pathogen-associated molecular patterns (PAMPs) are detected by pattern recognition receptors (PRRs) [9,10]. Toll-like receptors (TLRs) are the most widespread class of membrane-bound innate immune receptors, responsible for PAMPs recognition, such as lipopolysaccharides (LPS) [11]. Besides the role of PRRs, TLRs also play a critical role in the production of immune effectors through the activation of intracellular signaling cascades [12]. According to an analysis of TLR repertoires in 85 metazoans, marine bivalves present the largest and most diversified repertoire of TLRs in the animal kingdom [13]. It was hypothesized that the expansion of the TLR gene family in mussels follows a functional diversity motivated by the biological features of these organisms and their habitats [13]. This highlighted the role of TLRs in the innate immune system of bivalves.

Toll-like receptor 4 (TLR4) is a major member of the TLR family and plays a fundamental role in pathogen recognition and the activation of innate immunity [12]. The stimulation of TLR4 by LPS has been intensively investigated, and the relatively conserved LPS/TLR4 signal transduction pathway has been characterized in various vertebrates [11,14]. TLR4 signaling has generally been divided into myeloid differentiation primary response gene 88 (MyD88)-dependent and MyD88-independent pathways, responsible for proinflammatory cytokine expression and the induction of Type I interferons and interferon-inducible genes, respectively [15,16,17]. Nowadays, increasing evidence suggests that bivalves have a TLR4-MyD88 pathway, although the TLRs of these animals are not orthologous to those of vertebrates [18]. This pathway in bivalves is also supposed to be critical against bacterial infection and/or environmental stress, though it is not well studied in invertebrates. For example, Xing et al. [19] systematically characterized 18 *TLR* genes in Yesso scallops (*Patinopecten yessoensis*) and confirmed their inducible expression patterns under acidifying exposure, including one *TLR4* gene. Recent transcriptome analysis indicated the involvement of the TLR4-MyD88 pathway in the immune response of *Crassostrea hongkongensis* against *Vibrio parahemolyticus* [20]. Subsequent RNA interference (RNAi) and Vibrio challenge assay provided a novel insight into TLR4 in the innate immune response of *C. hongkongensis* [21]. According to a comparative proteomics and transcriptomics analysis, the *TLR4* gene also played a vital role in innate immunity induction when pearl oyster *Pinctada fucata martensii* was implanted with a spherical nucleus, which is a common process in artificial pearl production [22]. As for TLR4 signaling in bivalves under heat stress, this still needs to be investigated.

Previous transcriptome analysis of F3 generation of “Hongmo No. 1” scallops revealed that the expression of numerous genes involved in the immune system was affected under heat stress, including genes in the TLR4 signaling pathway (unpublished data). The aim of this study was to further investigate the TLR4 signaling pathway under heat stress. The full-length transcripts of *TLR4*, *MyD88* and the TNF receptor-associated factor 6 gene (*TRAF6*) in “Hongmo No. 1” (*HMTLR4*, *HMMyD88* and *HMTRAF6*) were discovered using RACE, and the expression patterns of four genes in the TLR4-MyD88 signaling pathway were also detected when scallops were exposed to heat stress and/or LPS stimulation. Furthermore, *HMTLR4* was silenced by RNA interference in vivo, followed by transfer to an environment with a temperature of 30 °C. Then, the expression of downstream genes and immunological indicators was analyzed to investigate the role of *HMTLR4* in the immune response to heat stress. This work will enrich the knowledge of scallop immunity challenged upon exposure to heat stress, with the aim of improving the health of aquaculture and resistance to disease and abiotic stress.

## 2. Materials and Methods

### 2.1. Animal Information

All scallops used in this study were from the sixth generation of scallop strain “Hongmo No. 1”. In total, four hundred three-month-old scallops (shell length 40~50 mm) were collected from a local scallop farm in Zhanjiang, Guangdong province, China, in September 2022. Before treatment, they were acclimated over a week in 28–30‰ filtered seawater (FSW) at 23–24 °C. During acclimation, the scallops were not fed.

### 2.2. Heat Stress Treatment

After acclimation, some scallops were maintained in 24 °C FSW as a control and others were transferred to 26 °C, 28 °C, 30 °C and 32 °C FSW, respectively. Each treatment consisted of three replicates, with 15 scallops per replicate. Two specimens were taken randomly from each replication tank at 6, 12, 24, 48 and 72 h post-transfer. Part of the shell edges (about 10 mm long and 5 mm wide) of each scallop was chipped away and hemolymph was taken from the adductor muscle of the shell through the notch with a syringe. Then, it was centrifuged at 800× *g* for 10 min at 4 °C. After removing the supernatant liquid, 1 mL Trizol (Thermo Fisher Scientific, Waltham, MA, USA) was added, and the hemolymph sample was stored at −80 °C. For the control group, the gill, adductor muscle and mantle of each scallop were then dissected immediately. For the treatment group, the gill tissue of each scallop was sampled. Then, tissue samples were frozen immediately using liquid nitrogen and stored at −80 °C.

### 2.3. Injection of Lipopolysaccharide (LPS) at 24 °C and 32 °C

For this experiment, acclimated scallops were divided randomly into four groups, with 40 scallops per group. One hundred microliters of LPS (10 μg/mL) from *Escherichia coli* 0111:B4 (Sigma–Aldrich, St. Louis, MO, USA) were injected into the adductor muscle of each scallop in two of the groups. Those in the other two groups were injected with 100 μL PBS (0.01 M, pH 7.4). After injection, two groups of scallops injected with LPS were transferred to 24 °C and 32 °C FSW, respectively. Those injected with PBS were treated in the same way. Six scallops were randomly sampled from each group at 3, 6, 12, 24 and 48 h post-transfer. All tissue samples were obtained and stored as described above.

### 2.4. RNA Extraction

The total RNA of the scallop hemolymph and tissues was extracted using RNAiso Plus (Takara, Kusatsu, Japan) according to the manufacturer’s protocol. The integrity of total RNA was analyzed using RNase-free agarose gel electrophoresis (1.5% agarose gel). The concentration and purity of the total RNA were assessed using SimpliNano (Biochrom, Berlin, Germany).

### 2.5. Molecular Cloning and Sequencing

The total RNA of scallop gill tissues was used as the template for RACE library construction and first-strand cDNA was produced by the SMARTer^®^ RACE 5′/3′ Kit (Clontech, San Jose, CA, USA). Then, 5′- and 3′-RACE PCRs were performed with the primer sets in Table 1 and the PCR products were purified. The target fragment was introduced into the pEAZY^®^-Blunt Zero vector (TransGen Biotech, Beijing, China) and connected at 25 °C for 20 min. The ligating system was as follows: 1 µL pEAZY^®^-Blunt Zero vector and 4 µL PCR product. The ligation products were transformed into *E. coli* DH5α competent cells by the heat shock method and an appropriate amount of bacterial liquid was pipetted and evenly coated on an LB ampicillin resistance solid medium. Ten different independent positive clones were picked for each RACE insert and then sequenced by Sangon Biotech Co., Ltd. (Shanghai, China).

### 2.6. Bioinformatic Analysis

The complete cDNAs detected by RACE were analyzed by the NCBI ORF Finder (https://www.ncbi.nlm.nih.gov/orffinder/) (accessed on 24 March 2023) for the gene open reading frame (ORF) and deduced amino acid sequences. Then, the isoelectric point and molecular weight were predicted by Expasy pI/Mw (https://www.expasy.org/resoures/compute-pi-mw) (accessed on 21 March 2023) and the conserved domains were analyzed according to the protein information resource SMART database (https://smart.embl.de/) (accessed on 24 March 2023). Full-length protein sequences of orthologs were downloaded from NCBI and multiple sequence alignments were performed using ClustalX with default parameters. After manual adjustments, maximum likelihood trees were constructed using MEGA 7.0.26 with the 1000 bootstrap repetitions [23,24,25].

### 2.7. Real-Time Quantitative PCR (RT-PCR)

The total RNA was used for first-strand cDNA synthesis. First-strand cDNA was prepared with six biological replicates and synthesized from 1 μg total RNA by the Prime Script TM RT reagent Kit with the gDNA Eraser (Takara, Kusatsu, Japan). The RT-PCR was performed with the primers in Table 1 on the LightCycler 96 real-time PCR instrument (Roche, Welwyn, UK) using the SYBR Green real-time PCR Kit (Takara, Kusatsu, Japan). *β-actin* gene expression was used to normalize gene expression. Cycling parameters, amplification efficiency analysis, standard curve construction and control settings of the microplate were the same as those in our previous study on *Crassostrea gigas* [26]. Relative gene expression levels were calculated by the 2^−(ΔΔCt)^ method [27] and then data were analyzed by one-way analysis of variance (one-way ANOVA) with a significant threshold of *p* < 0.05.

### 2.8. RNA Interference (RNAi) of HMTLR4 Gene Expression In Vivo

A partial cDNA fragment of *HMTLR4* was amplified using the primer pair TLR4-sense and TLR4-antisense, listed in Table 1. The purified PCR product was ligated into the pEAZY^®^-Blunt Zero vector (TransGen Biotech, Beijing, China) and, after being transferred, the positive clones were selected for Sanger sequencing. The plasmids containing either the forward insert or the reverse insert were chosen and linearized separately by the Pst I enzyme (New England Biolabs, Hitchin, UK). The purified products were used as the template for in vitro transcription and the double-strand RNA (dsRNA) was synthesized using the T7 RNAi Transcription Kit (Vazyme, Nanjing, China) following the manufacturer’s instructions.

After acclimating at 24 °C for one week, scallops were randomly divided into four groups. One hundred microliters of dsRNA (1 μg/μL) was injected into the adductor muscle of each scallop in the two groups. Those in the other two groups were injected with 100 μL PBS (0.01 M, pH 7.4) as a control. After injection with dsRNA, two groups of scallops were transferred to FSW at temperatures of 24 °C and 32 °C, respectively. Scallops injected with PBS were treated in the same manner. Six scallops were randomly sampled from each group at 24, 48, 72 and 96 h post-injection. All tissue samples were obtained and stored as described above.

### 2.9. Enzyme-Linked Immunosorbent Assay (ELISA)

Sandwich-type ELISAs were performed for the detection of tumor necrosis factor alpha (TNF-α) and lysozyme (LZM) in scallop gills using the commercially available kits for tilapia (*Oreochromis niloticus*) (Shanghai Enzyme-linked Biotechnology Co., Ltd., Shanghai, China). Antibodies against TNF-α and LZM of tilapia were used to pre-coat the micro-wells in the two ELISA kits, respectively. Reconstituted TNF-α and LZM of tilapia were used as the standard samples. Briefly, tissues were homogenized in 0.01 M PBS and then centrifuged at 1000× *g* for 10 min. Fifty microliters of standard substance and sample supernatant were added to micro-wells that were coated with polyclonal rabbit anti-TNF-α or anti-LZM. Each well was spiked with 100 μL of HRP-labeled antibody and then incubated at 37 °C for 1 h. After washing, substrate solution was added and incubated at 37 °C for 15 min. When the reaction was terminated, the average optical density of each well was detected by the Synergy H1 Multi-Mode Microplate Reader (BioTek, Winooski, VT, USA) at a 450 nm wavelength and the content was determined by near regression equations based on gradient-diluted standard chemicals. The data were analyzed by one-way ANOVA with a significant threshold of *p* < 0.05.

## 3. Results

### 3.1. Gene Cloning and Sequence Analysis of HMTLR4, HMMyD88 and HMTRAF6

The transcript of *HMTLR4* was obtained by the RACE method, and it was 3016 bp in length, containing the open-reading frame (ORF) of 2643 bp, 5′-untranslated regions (UTR) of 311 bp and 3′-UTR of 63 bp (Figure 1A). It was predicted to encode a protein of 880 amino acids (aa) with a molecular weight (MW) of 101.306 KDa and isoelectric point (pI) of 5.76. The molecular structure prediction for the deduced amino acid sequence revealed one signal peptide (1-24 aa), one LRR-NT conserved domain (56-92 aa), seven LRR conserved domains (109-128 aa, 134-153 aa, 243-265 aa, 326-348 aa, 350-365 aa, 559-577 aa, 582-601 aa), one LRR-TYP conserved domain (535-558 aa), one membrane-spanning domain (681-703 a) and one TIR conserved domain (738-880 aa) (Figure 1A). When compared with orthologs in several other bivalves, TIR was the most conserved domain, and the predicted TLR4 protein in hybrid scallop “Hongmo No. 1” showed the closest homology to that predicted in *Mizuhopecten yessoensis* (67.3%) (Appendix A). The full sequence of the *HMTLR4* transcript was submitted to GenBank under accession number OQ750685.

The transcript of *HMMyD88* (GenBank accession number OQ750686) was 2393 bp, which contained 5′-UTR of 914 bp, ORF of 1062 bp and 3′ UTR of 417 bp (Figure 1B). *HMMyD88* was predicted to encode a protein of 353 aa with a MW of 42.408 kDa and pI of 6.80, including a death domain (30-118 aa) and a TIR domain (164-299 aa), as shown in Figure 1B. The two predicted domains of *MyD88* were highly conserved in bivalves, and the predicted MyD88 protein in “Hongmo No. 1” showed the closest homology to that in *Argopecten irradians* (99.7%) (Appendix A). The transcript of *HMTRAF6* (GenBank accession number OQ750687) was 3254 bp, which contained 5′-UTR of 312 bp, ORF of 2046 bp and 3′ UTR of 896 bp (Figure 1C). It was predicted to encode a protein of 681 aa with a MW of 77.269 kDa and pI of 6.10. The molecular structure prediction for the deduced amino-acid sequence revealed one RING domain (123-163 aa), two zinc finger domains (208-262 aa and 262-320 aa), a coiled-coil region (480-525 aa) and a MATH domain (532-659 aa) (Figure 1C). The predicted TRAF6 protein in “Hongmo No. 1” showed the closest homology to that in *Pecten maximus* (89.3%) (Appendix A).

### 3.2. Phylogenetic Analysis of HMTLR4, HMMyD88 and HMTRAF6

Phylogenetic analysis based on the deduced amino acid sequence was performed to investigate the relationship between the above three genes and their gene family members. As shown in Figure 1D, the TLR4 proteins of Mollusca formed a major group distinct from those of vertebrates, including mammals, aves and teleosts. Within the Mollusca clade, the predicted protein encoded by *HMTLR4* firstly clustered with TLR4 from scallops (*M. yessoensis* and *P. maximus*) and then clustered with TLR4 from oysters, mussels and abalones (Figure 1D). Similar phylogenetic relationships were also found in *HMMyD88* and *HMTRAF6* (Figure 1E,F).

### 3.3. Expression of HMTLR4, HMMyD88, HMIRAK4 and HMTRAF6 at Different Temperatures

Besides the three genes above, the expression patterns of another gene in the TLR4-MyD88 pathway, interleukin receptor associated kinase 4 (*HMIRAK4*), were also investigated. For a clear understanding about the tissue specificity of gene expression, the expression of four genes in the TLR4-MyD88 pathway were detected in the gill, adductor muscle, mantle and hemolymph of scallops sampled from the control group (24 °C) at 0 h. The relative expression of *HMTLR4* in the mantle was the highest among the four kinds of tissues, and the expression of *HMTLR4* in the gill was also significantly higher than that in the adductor muscle and hemolymph (Appendix A). As for *HMMyD88*, *HMIRAK4* and *HMTRAF6*, they showed similar expression patterns, and their transcripts in gill tissue were the most abundant (Appendix A).

The expression patterns of the above four genes under heat stress were primarily investigated in scallop gill tissues for their key roles in responses to environmental changes. At 6 h after transfer, *HMTLR4* expression was higher in the gills of scallops transferred to 26 °C FSW but decreased significantly in those transferred to 28 °C, 30 °C and 32 °C FSW (Figure 2A). At 12 h, *HMTLR4* expression exhibited no significant difference in samples from 24 °C, 26 °C and 30 °C groups and a relative low level in those from 28 °C and 32 °C groups (Figure 2A). There was no significant difference in *HMTLR4* expression between all groups at 24 h. The expression patterns of *HMTLR4* at 48 h were similar to those at 6 h, except for an insignificant decrease in those exposed to 28 °C FSW (Figure 2A). At 72 h, *HMTLR4* expression in samples exposed to 26 °C FSW remained at a relatively high level, but, interestingly, *HMTLR4* expression in 32 °C groups increased slightly when compared to those in 24 °C groups (Figure 2A). As for the expression of *HMMyD88* at 6 h, it was similar to the expression pattern of *HMTLR4*, except for an insignificant decrease in samples exposed to 28 °C and 30 °C (Figure 2B). The expression of *HMMyD88* in gills sampled from the control group was significantly higher than in others at 12 h, and there was no significant difference in *HMMyD88* expression between all groups at 24 h (Figure 2B). At 48 h and 72 h, *HMMyD88* expression in samples exposed to 26 °C FSW was relatively high, and there was no significant difference between *HMMyD88* expression in gill samples from 24 °C and 28 °C groups (Figure 2B). It should be noted that *HMMyD88* expression in scallops exposed to 32 °C remained at the lowest level within 72 h. *HMIRAK4* and *HMTRAF6* showed similar expression patterns in scallops under heat stress. As shown in Figure 2C,D, their expression in scallops exposed to 26 °C FSW was significantly higher than that of others at 6 h after transfer, but, at 12 h, gene expression in samples from the control group was at the highest level. At 48 h, the expression patterns of *HMIRAK4* and *HMTRAF6* were similar to that of *HMTLR4* (Figure 2C,D). There was no significant difference in gene expression between all groups at 72 h (Figure 2C,D).

### 3.4. The Expression of Genes in the TLR4-MyD88 Pathway after LPS Injection at 24 °C and 32 °C

The response of four TLR4 pathway genes to LPS stimulation was investigated at different temperatures. At 24 °C, the expression of *HMTLR4* increased significantly 3 h after injection with LPS and reached the highest level at 6 h (Figure 3A). Then, the expression of *HMTLR4* in scallops injected with LPS decreased to the same level as those injected with PBS at 24 h (Figure 3A). When transferred to 32 °C FSW after injection, the expression of *HMTLR4* showed no significant difference between samples injected with LPS and those injected with PBS (Figure 3B). *HMMyD88* presented a different expression pattern than that of *HMTLR4*. At 24 °C, a significant increase in *HMMyD88* could also be found 3 h after injection with LPS, and the relatively higher level under LPS stimulation persisted until 24 h (Figure 3C). When transferred to 32 °C FSW after injection, the expression of *HMMyD88* in samples injected with LPS was relatively higher than those injected with PBS within 6 h, but a smaller difference was shown when compared with those at 24 °C (Figure 3D). *HMIRAK4* and *HMTRAF6* showed similar expression patterns in response to LPS stimulation. Their expression increased significantly 6 h after injection with LPS, whether at 24 °C or at 32 °C, and the time was shorter in maintaining a higher level of gene expression at 32 °C than that at 24 °C (Figure 3E–H).

### 3.5. Levels of TNF-α and LZM under Different Stress Conditions

Levels of TNF-α and LZM were preliminarily investigated in the gill tissue of scallops under heat stress, and partial samples exposed to 32 °C FSW were selected for comparison with those in 24 °C FSW. As shown in Appendix A, the concentrations of TNF-α were not influenced by the increase in the environmental temperature at 6 h and 48 h, although the three TLR4 pathway-related genes in this study showed significant changes in abundance. Whereas, the activity of LZM increased significantly at 48 h after being transferred to 32 °C FSW (Appendix A). Levels of TNF-α and LZM were also detected in the gill tissue of scallops injected with LPS, and they increased significantly at 6 h and 48 h when compared with those injected with PBS, whether at 24 °C or at 32 °C (Table 2). In addition, no obvious impact of heat stress could be observed because both TNF-α and LZM showed similar levels between samples injected with LPS and PBS at different temperatures (Table 2).

### 3.6. The Effect of HMTLR4 Suppression on the Expression of Downstream Genes and the Levels of TNF-α and LZM

The expression of *HMTLR4* was primarily investigated in the gills of scallops injected with PBS and dsRNA at 24 °C. As shown in Figure 4A, the expression of *HMTLR4* in gills decreased significantly at 24 h but increased at 48 h after injection with dsRNA. Then, it was reduced to the same level as the expression of *HMTLR4* in samples injected with PBS (Figure 4A). Similar suppression of *HMTLR4* could be observed in scallops injected with dsRNA that underwent immediate transfer to 32 °C FSW, except for an insignificant difference at 48 h and a relatively higher level in the control group at 96 h (Figure 4B).

Then, we tested the effect of *HMTLR4* suppression on the expression of its downstream genes, *HMMyD88*, *HMIRAK4* and *HMTRAF6*. At 24 °C, *HMMyD88* did not show significant differences in transcript abundance until 96 h after injection with dsRNA targeting *HMTLR4* (Figure 4C). By contrast, when scallops were injected with dsRNA targeting *HMTLR4* at 32 °C, the expression of *HMMyD88* was significantly lower than that in the control group, except at 48 h after injection (Figure 4D). When *HMTLR4* was silenced at 24 °C, *HMIRAK4* showed similar expression trends within 96 h (Figure 4E). But, at 32 °C, a significant decrease appeared later at 48 h after injection (Figure 4F). As for the expression of *HMTRAF6,* it decreased immediately at 24 h but showed a higher level than that in the control group at 72 h after injection with dsRNA at 24 °C (Figure 4G). At 32 °C, there was no significant difference in the expression of *HMTRAF6* between the PBS and dsRNA treatment groups within 72 h, and a lower expression level of *HMTRAF6* in scallops injected with dsRNA appeared at 96 h (Figure 4H).

Scallops injected with dsRNA and PBS were selected, and the levels of TNF-α and LZM in gill tissues were determined by ELISA assay. At 48 h, TNF-α presented significantly lower concentrations in scallops injected with dsRNA when compared with those injected with PBS at 24 °C (Figure 5A). At 32 °C, a similar decrease in TNF-α was found in scallops injected with dsRNA, but the level was restored at 96 h (Figure 5A). LZM showed similar concentration variation trends when scallops were injected with dsRNA targeting *HMTLR4*, whether at 24 °C or at 32 °C (Figure 5B). Interestingly, the LZM of samples injected with dsRNA showed relatively higher levels than those of samples injected with PBS at 96 h after being transferred to 32 °C FSW (Figure 5B).

## 4. Discussion

Acute heat stress poses significant challenges to the immune system of marine bivalves, which are particularly vulnerable to extremely high temperatures. Recent advancements in molecular biotechnologies, such as transcriptome sequencing, have highlighted the importance of signal transduction pathways in understanding this process. In this study, we identified transcripts of three crucial genes involved in the TLR4-MyD88 signaling pathway in a novel scallop strain “Hongmo No. 1”. Our aim was to investigate the role of this specific signal transduction pathway in the immune response of scallops to acute heat stress.

Three genes, *HMTLR4*, *HMMyD88* and *HMTRAF6*, in the TLR4-MyD88 signaling pathway were identified in “Hongmo No. 1” and predicted to encode proteins with typical function domains. Seven LRR domains and one TIR domain, responsible for ligand recognition and the recruitment of signaling adapters, respectively [12,28], were presented in the deduced amino acids of *HMTLR4*. There was obvious variation in the number of LRR domains and TLR4 proteins among species, but, generally, more LRRs were present in the TLR4 proteins of invertebrates [29]. This supported the idea that TLR genes in both vertebrates and invertebrates are the result of independent gene family expansion and loss events. Such evolution of TLRs may be attributed to a “response to a changing array of binding requirements” [18,30,31,32]. MyD88 was the critical downstream adapter on the MyD88-dependent signaling pathway that interacted with TLR4 through the TIR domain [15,33]. In addition to the TIR domain, MyD88 also contains a death domain by which it recruits and activates a death domain-containing kinase, IRAK4 [14]. We found high conservation in the putative protein of *HMMyD88* with typical TIR and death domains but failed to obtain the transcript of its downstream IRAK4 coding gene. Fortunately, a homolog of the adaptor protein TRAF6 coding gene, critical for IRAK4, was identified in “Hongmo No. 1” scallops, and showed high similarity in deduced amino acids with homologs in other species. In our opinion, the identification of *HMTLR4*, *HMMyD88* and *HMTRAF6* supported the view that bivalves have a TLR- and MyD88-based immune signaling pathway [19,20,21,22,34].

Then, the expression patterns of four genes in the TLR4-MyD88 signaling pathway, including *HMIRAK4*, were investigated in “Hongmo No. 1” scallops at different temperatures for a global view of their transcriptional response to heat stress. Generally, these genes were downregulated when scallops were exposed to extremely high temperatures, especially those over 30 °C. This is different from the heat stress-mediated activation of the TLR signaling pathway in most homeothermic organisms, such as Bama miniature pigs [35], which is supposed to trigger the release of cytokines and coordinate a proinflammatory response [36,37,38]. But, similar suppression of the TLR4-MyD88 pathway can also be found in both homeothermic and poikilothermal animals under acute heat stress. For example, the TLR4 coding gene was significantly downregulated in tissues of Indian major carp catla *Catla catla* at temperatures over 30 °C [39]. These results suggest an immediate and direct effect of heat on TLR pathway activation; however, conclusive results and the significance of these findings in the pathogenesis of heat-stressed animals have not been clearly defined [38]. In addition, there was no significant difference in the level of TNF-α between “Hongmo No. 1” scallops at 24 °C and 32 °C FSW within 48 h, although genes in the TLR4-MyD88 pathway were significantly downregulated at 32 °C. Thus, this requires more evidence to confirm whether the TLR4-MyD88 signaling pathway affects innate immunity in scallops.

The increase in immune parameters at 24 °C was preceded by an increase in gene expression, which may indicate a relationship. This was consistent with the role of TLR4 as an important sensor for LPS to induce the release of critical proinflammatory cytokines that are necessary to activate potent immune responses [11,40]. Under acute heat stress, the LPS-induced TLR4-MyD88 signaling pathway in scallops was obviously suppressed, which was revealed by the smaller increase in gene expression. This supported the negative effect of heat on TLR pathway activation, as shown in the initial heat stress treatment experiment, where the decline was visible at 28 °C. A previous in vitro study on innate memory CD8+ T cells of mice indicated that heat shock protein (HSP) 70 could down-regulate TLR4 [41], and this provided a clue to the activation mechanism of the TLR4-MyD88 signaling pathway in scallops under heat stress. The *TLR4* genes in invertebrates are quite different from those in mammals, and it remains to be further investigated whether the activation of TLR4-MyD88 under heat stress is relatively conserved between vertebrates and invertebrates. Moreover, TNF-α and LZM were not significantly affected by heat stress and showed an obvious increase when injected with LPS, both at 24 °C and at 32 °C. This may indicate that adequate signal molecules were transduced for the activation of innate immunity in scallops, although the TLR4-MyD88 signaling pathway was negatively affected.

Subsequent RNAi targeting *HMTLR4* provided more information on the TLR4-MyD88 signaling pathway in the immune response of “Hongmo No. 1” scallops. *HMTLR4* silencing, to a greater or lesser extent, suppressed the expression of *HMMyD88*, *HMIRAK4* and *HMTRAF6*, whether at 24 °C or at 32 °C. Previous studies on several bivalves, such as *Chlamys farreri* and *C. hongkongensis*, also indicated that the inhibition of the *TLR* gene induced a decrease in its downstream mediator molecules [21,42], demonstrating the key role of *TLR* in the activation of TLR- and MyD88-based signaling pathways in bivalves [43]. TNF-α and LZM showed lower levels at different time points after injection with dsRNA targeting *HMTLR4*, which indicated that the block of TLR4-MyD88 signal transduction negatively affected the innate immunity of scallops. According to the gene expression at 32 °C, acute heat stress seemed to weaken or strengthen the response of downstream mediator molecules to the inhibition of *HMTLR4*. This led us to a hypothesis that heat stress could directly influence the downstream targets of *HMTLR4*, but more convincing evidence needs to be provided. Investigating the potential involvement of heat shock proteins, such as HSP 70, in downstream signaling events could be a promising area for further study on the TLR4-MyD88 pathway of bivalves. Such research could deepen our understanding of the complex molecular mechanisms underlying the immune response to acute heat stress in marine bivalves and enhance our ability to develop effective strategies for mitigating the impacts of extreme temperature events on these aquaculture organisms.

## 5. Conclusions

In this study, *HMTLR4*, *HMMyD88* and *HMTRAF6* were characterized from a new scallop strain, “Hongmo No. 1” (Bohai Red ♂ × *A. irradians concentricus* ♀); then, the expression of four genes in the TLR4-MyD88 signaling pathway, including *HMIRAK4*, was investigated in scallops under heat stress and/or LPS stimulation. Acute heat stress obviously inhibited the TLR4-MyD88 signaling pathway, even with LPS stimulation, but innate immunity in scallops was activated successfully according to the levels of proinflammatory cytokines and immune enzymes. RNAi targeting *HMTLR4* at different temperatures indicated the key role of TLR4-MyD88 signal transduction in the immune response of “Hongmo No. 1” scallops. Acute heat stress seemed to affect the response of downstream mediator molecules to the inhibition of *HMTLR4*. These results provide more information about bivalve immunity under heat stress and can be beneficial for the genetic improvement of heat tolerance to cope with more frequent and extreme heat wave events.

## Figures and Tables

**Figure 1 animals-14-00497-f001:**
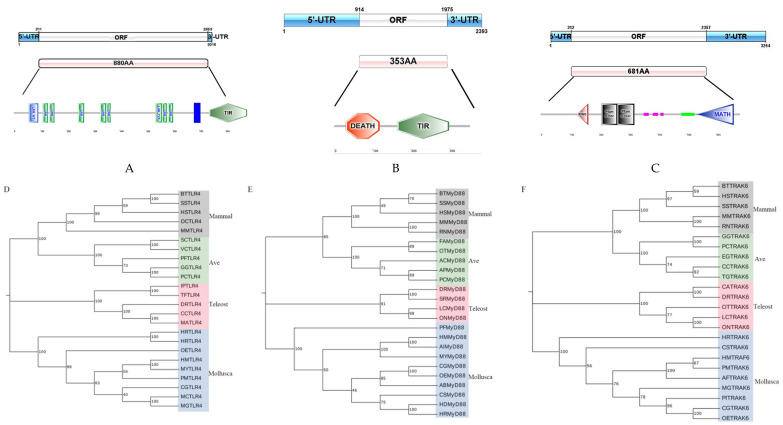
*HMTLR4*, *HMMyD88* and *HMTRAF6* genes in scallop “Hongmo No. 1” and phylogenetic trees of their putative proteins. (**A**–**C**) Schematic presentation of *HMTLR4*, *HMMyD88* and *HMTRAF6*, respectively. Sky-blue and white boxes show transcript UTRs and their ORF. Pink box show putative proteins. In predicted amino acid sequences, conserved domains were labeled with texts and different colors. (**D**–**F**) Phylogenetic trees based on the predicted amino acid sequences of *HMTLR4*, *HMMyD88* and *HMTRAF6*, respectively. Species names are abbreviated as BT for *Bos taurus*, SS for *Sus scrofa*, HS for *Homo sapiens*, OC for *Oryctolagus cuniculus*, MM for *Mus musculus*, RN for *Rattus norvegicus*, SC for *Serinus canaria*, VC for *Vidua chalybeata*, PF for *Patagioenas fasciata monilis*, GG for *Gallus gallus*, PC for *Phasianus colchicus*, FA for *Ficedula albicollis*, OT for *Onychostruthus taczanowskii*, AC for *Acanthisitta chloris*, AP for *Anas platyrhynchos*, EG for *Egretta garzetta*, CC for *Cyanistes caeruleus*, TG for *Taeniopygia guttata*, IP for *Ictalurus punctatus*, TF for *Tachysurus fulvidraco*, DR for *Danio rerio*, CC for *Cyprinus carpio*, MA for *Megalobrama amblycephala*, SR for *Sinocyclocheilus rhinocerous*, LC for *Larimichthys crocea*, ON for *Oreochromis niloticus*, CA for *Carassius auratus*, OT for *Oncorhynchus tshawytscha*, PF for *Pinctada fucata*, AI for *Argopecten irradians*, AB for *Anadara broughtonii*, HD for *Haliotis diversicolor*, CS for *Cyclina sinensis*, HR for *Haliotis rufescens*, OE for *Ostrea edulis*, MY for *Mizuhopecten yessoensis*, PM for *Pecten maximus*, CG for *Crassostrea gigas*, MC for *Mytilus californianus*, MG for *Mytilus galloprovincialis*, AF for *Azumapecten farreri* and PI for *Pinctada imbricata*.

**Figure 2 animals-14-00497-f002:**
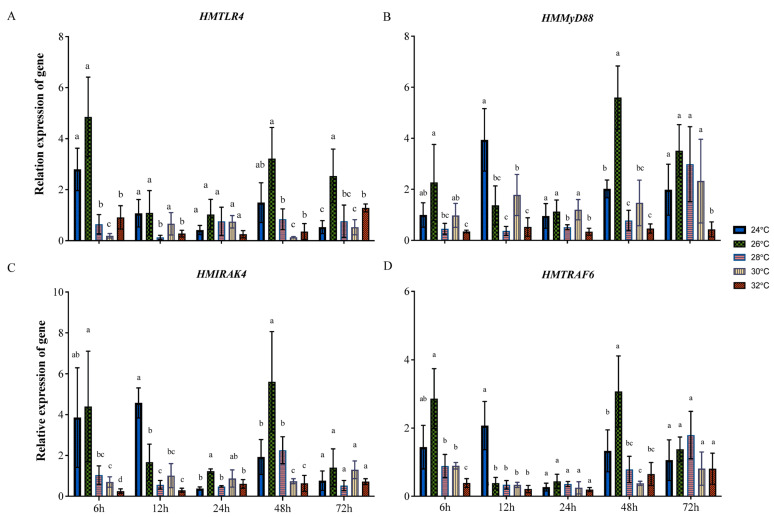
Expression profiles of *HMTLR4* (**A**), *HMMyD88* (**B**), *HMIRAK4* (**C**) and *HMTRAF6* (**D**) in gills of scallops at different temperatures. Bars represent standard deviation. For statistical analysis, a separate one-factor analysis of variance (ANOVA) was performed at each time point and levels were accepted as significant if *p* value < 0.05. The different letters indicate significant differences in gene expression between groups at the same time point.

**Figure 3 animals-14-00497-f003:**
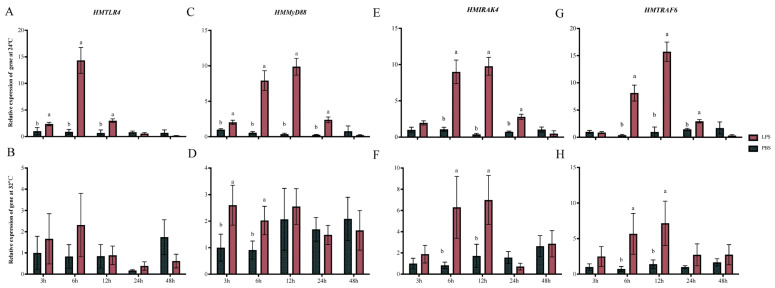
Expression profiles of *HMTLR4*, *HMMyD88*, *HMIRAK4* and *HMTRAF6* in gills of scallops under LPS stimulation at 24 °C and 32 °C. (**A**,**B**) Expression profiles of *HMTLR4* at 24 °C and 32 °C. (**C**,**D**) Expression profiles of *HMMyD88* at 24 °C and 32 °C. (**E**,**F**) Expression profiles of *HMIRAK4* at 24 °C and 32 °C. (**G**,**H**) Expression profiles of *HMTRAF6* at 24 °C and 32 °C. Bars represent standard deviation. For statistical analysis, a separate one-factor analysis of variance (ANOVA) was performed at each time point and levels were accepted as significant if *p* value < 0.05. The different letters indicate significant differences in gene expression between groups at the same time point.

**Figure 4 animals-14-00497-f004:**
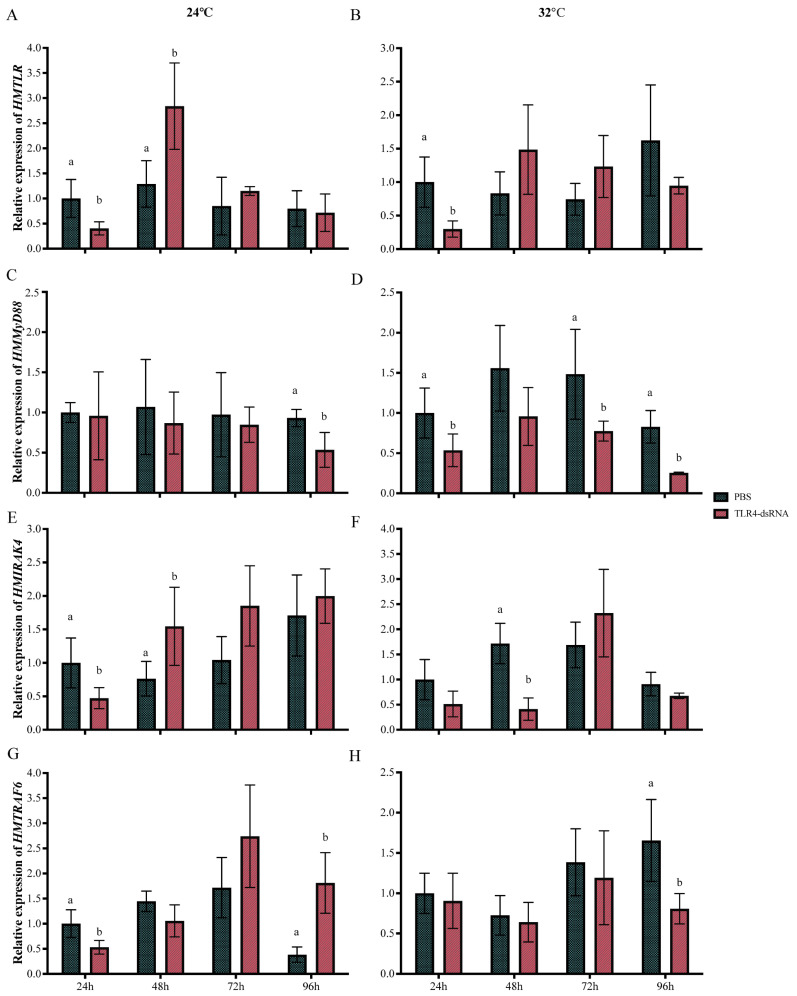
Expression profiles of *HMTLR4* (**A**,**B**),*HMMyD88* (**C**,**D**), *HMIRAK4* (**E**,**F**) and *HMTRAF6* (**G**,**H**) in gills of scallops injected with dsRNA at 24 °C and 32 °C. Bars represent standard deviations. Statistical analyses were carried out by one-way ANOVA and levels were accepted as significant if *p* value < 0.05. The different letters indicate significant differences in gene expression between groups at the same time points.

**Figure 5 animals-14-00497-f005:**
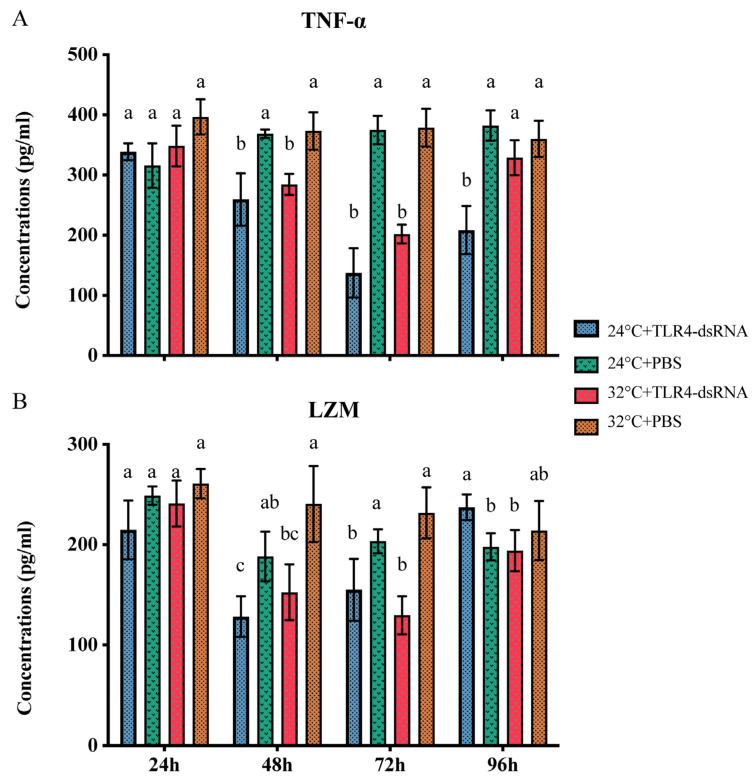
Levels of TNF-α (**A**) and LZM (**B**) in gills of scallops injected with dsRNA targeting *HMTLR4* at 24 °C and 32 °C. Bars represent standard deviations. Statistical analyses were carried out by one-way ANOVA and levels were accepted as significant if *p* value < 0.05. The different letters indicate significant differences between groups at the same time points.

**Table 1 animals-14-00497-t001:** Primers used in this study.

Primer	Sequence (5′–3′)	Application
*HMTLR4-5-1*	TGGAGACAGAGCCGTAACCCACG	5′RACE
*HMTLR4-5-2*	GGTGGAGACAGAGCCGTAACCCA	5′RACE
*HMMyD88-5-1*	AATTGTGGACACTATAGTTATGCTTGAC	5′RACE
*HMMyD88-5-2*	CATCACCGCCTTTAACCACAAT	5′RACE
*HMTRAF6-5-1*	GCCATTGCCTGTCTTTCAACCGT	5′RACE
*HMTLR4-3-1*	TAGATTTGTCGGGTGAAGTCGGTAGC	3′RACE
*HMTLR4-3-2*	TTCACGCTGGGAAGGTATCTGCT	3′RACE
*HMMyD88-3-1*	TTACAGGGCAAGTTGGGGTGT	3′RACE
*HMMyD88-3-2*	AAATTATGATGATTTCGCAGCTATG	3′RACE
*HMTRAF6-3-1*	GAAGGCTGTGATACGCAGGTGGT	3′RACE
*HMTLR4-F*	CCTTACCTGTATTTCCCCCC	RT-PCR
*HMTLR4-R*	GGTTTCTGACTGGACCCTTC	RT-PCR
*HMMyD88-F*	TTTTAGGAACAGGCTTTA	RT-PCR
*HMMyD88-R*	TGTGGACACGTTGATAAT	RT-PCR
*HMIRAK4-F*	TGTCGTACTGCTGCCTGAT	RT-PCR
*HMIRAK4-R*	TGGAAGCGTAGACTGGAGA	RT-PCR
*HMTRAF6-F*	TCAGGTGAAGCGATTGGG	RT-PCR
*HMTRAF6-R*	TCCTGGTGGAAAGGTAAT	RT-PCR
*β-actin-F*	CCGTGACTTGACCGATTACC	RT-PCR
*β-actin-R*	CCTTGATGTCCCTGACGATT	RT-PCR
*HMTLR4-sense*	GGTCAGTGTTTCGATTGGCG	DsRNA synthesis
*HMTLR4-anti*	CCCAAGTTGTTGTTGCTGATGA	DsRNA synthesis

**Table 2 animals-14-00497-t002:** Levels of TNF-α and LZM in scallops injected with LPS and PBS.

	Time (h)	Concentrations (pg/mL)			
		24 °C + LPS	24 °C + PBS	32 °C + LPS	32 °C + PBS
TNF-α	6	720.17 ± 95.03 *	368.70 ± 7.11	791.65 ± 71.13 ^†^	373.44 ± 30.98
	48	746.63 ± 57.24 *	382.12 ± 25.34	749.39 ± 58.73 ^†^	360.01 ± 30.10
LZM	6	367.00 ± 54.31 *	188.30 ± 24.78	424.84 ± 27.20 ^†^	240.54 ± 37.83
	48	387.73 ± 38.14 *	458.63 ± 36.42	458.63 ± 36.42 ^†^	214.01 ± 9.44

Values represent the mean value with standard deviation in six different individuals. Significant differences from the value in the control group at the same time points are represented by * and ^†^ at 24 °C and 32 °C, respectively (*p* < 0.05).

## Data Availability

The data presented in this study are available upon request from the corresponding author. The data are not publicly available due to the requirement of Guangdong Ocean University.

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
