# Peer review of "The Role of the TLR4-MyD88 Signaling Pathway in the Immune Response of the Selected Scallop Strain “Hongmo No. 1” to Heat Stress"

_animals, 2024, doi:10.3390/ani14030497_

Round 1

Reviewer 1 Report

Comments and Suggestions for Authors

Author Response

Thanks for your careful and professional review work on our paper. And your suggestions really mean a lot to us. Based on your comments, we have made extensive revision to our previous draft. The detailed point-to-point responses are listed below.

Introduction

  1. Line 48 and 50: The references have been corrected or provided as suggested.
  2. Line 64-68: We have tried to rewrite these sentences for easy reading. If any advice or comment, we would appreciate it.
  3. Line 70: Thanks for your suggestion about the reference, but we did not find an original article for such kind of summarized statement. After check, we decide to delete the previous reference because this well-known statement was so
  4. Line 74: We cited the original article that demonstrates the role of TLR4 in regulating LPS response.
  5. Line 76: “analyzation” has been changed to “analysis”.
  6. Line 83: The reference has been provided.
  7. Line 84: “conservative” has been changed to “conserved”.
  8. Line 85 and line 89: We cited the original articles as suggested.
  9. Line 94: Gene name has been written in italic type.
  10. Line 99-102, and line 109-110: These sentences have been rewritten. If any advice on better readability, we would appreciate it.

Material and methods

  1. Line 121: The number of scallops used in this study has been provided.
  2. Line 129-134: More details about hemolymph sampling were provided, including the storage method.
  3. Why control and treatment groups went under different tissue sampling?

Response: In this study, we mainly focused on the gene expression and immunity response of scallop gill tissues under different stress, and the tissues sampled from control group was used for the the tissue specific expression of target genes.

  1. Line 139-140: The information of LPS was provided.
  2. Line 142: What we referred to is the “each group with 40 scallops”, and the “sets” has been changed to “groups”. Additionally, the redundant “and” in sentences was deleted.
  3. Line 152-161: Details about the cloning were provided as suggested.
  4. Line 166-168 and 179: The corrections have been done and the missed information has been provided.
  5. Table 1 has been corrected as suggested.
  6. Line 197-201: These sentences have been edited. If any advice, would you please let us know?

Results

  1. Line 223, 226, 228 and 237: These sentences have been corrected as suggested.

Figure 1

  1. The figure 1 has been redrawn and “TRAK6” in Fig. 1F has been corrected as “TRAF6”. Additionally, the percentage of homology has been provided in line 230, 239 and 246.
  2. Line 267: “amino acid”has been corrected.
  3. Line 273: Please make this comment clearly? Would you mind pointing the problem with Figure 1D?

Figure 2

  1. Time points have been provided.

Figure 3

  1. We thought that you referred to the temperature (24℃ and 32℃), not the time points. Right? The temperature of 24℃ is the average temperature of seawater in the local scallop farm where we get these scallops in September 2022, whereas 32℃ was the highest temperature of surface seawater in this area. The heat stress experiment has proved that the four genes were inhibited when temperature was over 28℃. And in this experiment, we focus on the response of scallops to LPS stimulation at extreme high temperature, which was the main reason of setting two temperatures.

Figure 4

  1. This figure showed the gene expression in scallop gills after injected with dsRNA. According to many studies using dsRNA for RNA interference and our previous experience, the dsRNA usually gene silencing resulting from dsRNA usually can be assessed after 24 hours post-transfection, and the effect often last for days. This is why we did not detect the gene expression early but analyzed them more times after 48 h when compared with previous experiments.
  2. HMTLR has been changed to HTMLR4 in Figure 4.
  3. Line 378: The spelling of “until” has been corrected.
  4. Line 404-411, 418-421: These sentences have been edited for better readability.
  5. Line 423: The references have been corrected as suggested.
  6. Line 459-462 and 480-485: We tried to discuss more about the study and its application potential in a broader scientific view.

Reviewer 2 Report

Comments and Suggestions for Authors

This manuscript describes TLR4-Myd88 pathway gene expression in response to heat stress. The aim of the study is interesting and sound. However, there are a number of issues to be addressed for more clarity and accuracy.

1. The introduction should clearly explained why this study is focused on the Hongmo No. 1 line of this scallop species. In addition, the introduction could state how many TLRs do scallops possess for readers' reference.

The terminology 'strain' means the animal is genetically highly homogeneous. Are the animals used here actually such 'strain'?

Table 1: Correspondence between the primers and their applications are not clear. You can inset horizontal lines to make the primers into groups with respective applications.

Fig. 1. What is the difference between the two HMTLR4 in the tree?

Fig. 2: Labeling of Y-axis should be corrected. (from "Relation" to "Relative Expression Level")

Fig. 2, Fig. 3: Were ANOVA test for significant difference made between the groups within each time point?

Fig. 2 could be re-drawn in line graphs showing time course of the expression level for each temperature.

Fig. 3: Meaning of the panels A to H should be explained in the fig. legend.

L167,168: x cDNA Library --> 1st strand cDNA

L194: Detailed specification of the ELISA kits should be provided. Was the antibody against human TNF-alpha and lysozyme? It should be proven if these ELISA kit is applicable to scallop homologs. In addition, what were the standard sample for these kits?

L198: If these kit used mammalian TNF-alpha and lysozyme, the reactivities of the antibody must be substantially different from those against scallop proteins. Therefore, the concentration measurement of scallop proteins may not represent true value.

Table 2 contains too many effective numbers. The ELISA kits' accuracy will be less.

Comments on the Quality of English Language

The English usage looks fine to follow the contents, but should be mended moderately on a number of inappropriate expressions.

Author Response

Thanks for your careful and professional review work on our paper. And your suggestions really mean a lot to us. Based on your comments, we have made extensive revision to our previous draft. The detailed point-to-point responses are listed below.

Introduction

  1. Line 64-68: These sentences were edited for easy understanding about the aim of this study. This study focused on the “Hongmo No. 1” scallop strain because we want to obtain more information about further genetic improvement of this strain for heat tolerance.
  2. Line 93-95: We rewrote this sentence to state the number of TLR genes in Yesso scallop (Patinopecten yessoensis).
  3. Scallops used in this study are from the sixth generation of scallop strain “Hongmo No.1”, which means that they are genetically highly homogeneous as pointed.

Table 1

  1. Table 1 was modified for correspondence between the primers and their applications.

Figure 1

  1. We thought what you referred to are two “HRTLR4” in the previous version. Sorry for our mistake in drawing, we have changed them to “HRTLR4” and “HDTLR4”.

Figure 2

  1. Labeling of Y-axis have been corrected. Additionally, we tried to redraw it in line graphs, but found it unconcise. For better readability, we keep it in column graph.
  2. As pointed, we performed a separate one-way ANOVA within each time point, which was stated in the revised figure legend.

Figure 3

  1. Figure legend has been revised to provide the Meaning of the panels A to H and a clear statement for statical analysis.
  2. Line 175-176: “cDNA Library” has been changed to “first strand cDNA”as suggested.
  3. Line 205-207 and 210: The method for ELISA in this study was optimized based on an approach for ELISA that detected IL-1 and TNF-α in tilapia(Oreochromis niloticus). As suggested, we tried to provid more details about the ELISA experiment. We totally agreed with your opinion about the accuracy because there were no available antibodies for bivalve TNF-α and LZM, but wo do believe, to some extent, this could reflect the immune response of scallops. If any advice, we would appreciate it.